# Constrained Predictive Coding as a Biologically Plausible Model of the Cortical Hierarchy

**Siavash Golkar**[*1]          **Tiberiu Tesileanu**[*1]          **Yanis Bahroun**[1]

**Anirvan M. Sengupta**[2,3,4]          **Dmitri B. Chklovskii**[1,5]

[1] Center for Computational Neuroscience, Flatiron Institute
[2] Center for Computational Mathematics, Flatiron Institute
[3] Center for Computational Quantum Physics, Flatiron Institute
[4] Department of Physics and Astronomy, Rutgers University
[5] Neuroscience Institute, NYU Medical Center

{sgolkar,ttesileanu,ybahroun,dchklovskii}@flatironinstitute.org
anirvans.physics@gmail.com

## Abstract

Predictive coding (PC) has emerged as an influential normative model of neural computation with numerous extensions and applications. As such, much effort has been put into mapping PC faithfully onto the cortex, but there are issues that remain unresolved or controversial. In particular, current implementations often involve separate value and error neurons and require symmetric forward and backward weights across different brain regions. These features have not been experimentally confirmed. In this work, we show that the PC framework in the linear regime can be modified to map faithfully onto the cortical hierarchy in a manner compatible with empirical observations. By employing a disentangling-inspired constraint on hidden-layer neural activities, we derive an upper bound for the PC objective. Optimization of this upper bound leads to an algorithm that shows the same performance as the original objective and maps onto a biologically plausible network. The units of this network can be interpreted as multi-compartmental neurons with non-Hebbian learning rules, with a remarkable resemblance to recent experimental findings. There exist prior models which also capture these features, but they are phenomenological, while our work is a normative derivation. This allows us to determine which features are necessary for the functioning of the model. For instance, the network we derive does not involve one-to-one connectivity or signal multiplexing, which the phenomenological models require, indicating that these features are not necessary for learning in the cortex. The normative nature of our algorithm in the simplified linear case also allows us to prove interesting properties of the framework and analytically understand the computational role of our network's components. The parameters of our network have natural interpretations as physiological quantities in a multi-compartmental model of pyramidal neurons, providing a concrete link between PC and experimental measurements carried out in the cortex.

---

[*]Equal contribution

36th Conference on Neural Information Processing Systems (NeurIPS 2022).

# 1 Introduction

Over the past decades, predictive coding (PC), a normative framework for learning representations that maximize predictive power, has played an important role in computational neuroscience [1,2]. Initially proposed as an unsupervised learning paradigm in the retina [3], it has since been expanded to the supervised regime [4] with arbitrary graph topologies [5,6]. The PC framework has been analyzed in many contexts [7,8] and has found many applications, from clinical neuroscience [9,10] to memory storage and retrieval [11]. We refer the reader to [12,13] for recent reviews.

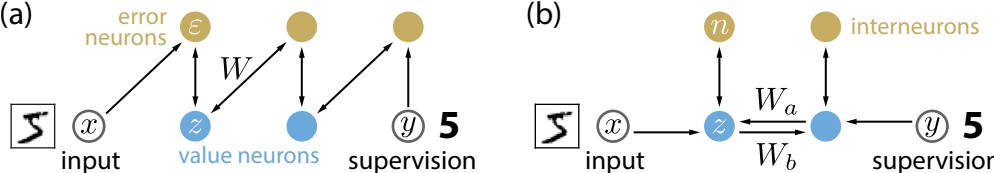

**Figure 1:** Schematic architecture of the predictive coding network (PCN) and our covariance-constrained network (BioCCPC). (a) PCN from [4], figure adapted from [13]. The intra-layer connectivity is to be one-to-one, while the inter-layer connectivity is symmetric. (b) Our BioCCPC network. There is no requirement for symmetric weights across layers, and the connectivity within layers can be diffuse.

Predictive coding is viewed as a possible theory of cortical computation, and many parallels have been drawn with the known neurophysiology of cortex [12]. While the initial works proposed a biologically plausible network [1,2], the connection with cortex was more closely examined in [14], where the PC module was mapped onto a cortical-column microcircuit. However, there are aspects of this mapping that have proved controversial. Among these are the requirement of multiple redundant cortical operations, the symmetric connectivity pattern, the one-to-one connectivity of value and error neurons, and also the requirement that feedback connections be inhibitory [12,15,16], as sketched in Fig. 1a . The presence of separate error and value neurons has itself been called into question [15].

The PC-based neural circuits also do not account for more recent experimental findings highlighting the details of computation in the cortex [17–26]. For example, the learning dynamics of multi-compartmental pyramidal neurons has been closely investigated. In particular, it was observed that the plasticity of the synapses of the basal compartment is driven by the activity in the apical tuft of the neuron by so-called calcium plateau potentials [19,20,24], leading to non-Hebbian learning rules [25]. These experiments have motivated the development of several models of microcircuits with multi-compartmental neurons [27–33]. In a number of cases, it has been shown that these models can replicate learning similar to the backpropagation algorithm under specific assumptions [30,34,35]. However, because of their rather phenomenological nature, detailed analysis of these models is in many cases challenging and one must resort to numerical simulations, rendering the task of understanding the role of various neurophysiological quantities difficult. To this date, a normative framework (PC or otherwise) that can explain these experimental findings is still lacking.

In this work, we show how to modify the PC framework to make it compatible with the aforementioned experimental observations while maintaining much of the analytical control afforded to use by a normative approach. Inspired by prior work which explored the effects of finding decorrelated representations [36–41], we add a decorrelating inequality constraint to the covariance of the latent representations of PC. Using this constraint, we derive an upper bound for the PC objective. By working in the linear regime, we can prove several useful properties of our algorithm. We show that the learning algorithm derived from this upper bound does not suffer from the issues of prior implementations and naturally maps onto the hierarchical structure of the cortical pyramidal neurons.

**Contributions**

- We introduce BioCCPC, a novel biologically plausible algorithm derived as a modification of predictive coding (PC), and show that it closely matches the known physiology of the cortical hierarchy.
- We find an analytic upper bound to the PC objective by imposing a decorrelation-inspired inequality constraint on the latent space.
- We interpret the different parameters of the algorithm in terms of the conductances and leaks of the separate compartments of the pyramidal neuron. We find that the neural

compartmental conductances encode the variances of the PC framework, and the somatic leak maps onto a thresholding mechanism of the associated eigenvalue problem.

## 2 Related work and review of predictive coding

The backpropagation algorithm [34, 42] is the predominant tool in the training of deep neural networks but is generally not considered biologically plausible [43]. Over the years, many authors have explored biologically plausible approximations or alternatives to backpropagation [30,31,44–58] (for a more complete review see [59]). These approaches generally fall into two categories. First are normative approaches, such as Predictive Coding [1–3], Target Propagation and variations [44,45,47], Equilibrium Propagation [49, 53] and others [51, 54–56], where one starts from a mathematically motivated optimization objective. These methods, by virtue of their normative derivation, have a firm grounding in theory; however, they do not fully conform to the experimental observations in the brain (see below and Sec. 4). The second approach is driven by biology, with network structures and learning rules inspired by experimental findings [28,30,31,48,60]. While these works mostly conform to the experimentally observed findings, they are more challenging to analyze because of their conjectured phenomenological nature.

The goal of the present paper is *not* to propose yet another biologically plausible alternative to backpropagation. It is rather to demonstrate that the normative framework of predictive coding, when combined with a constraint, can indeed closely match experimental observations. For this reason, in this work, we focus on comparing our method with previous implementations of predictive coding and do not concern ourselves with other biologically plausible alternatives to backpropagation. The relationship between the PC framework and backpropagation was explored in [4,61–64]. The advantages of PC over backpropagation were highlighted in [13]. For other work addressing the realism problems of PC see [65, 66].

We now turn to a brief review of the predictive coding framework from [4] before delving into the details of our method.

**Notation.** Bold upper case $\mathbf{M}$ and lower case variables $\mathbf{v}$ denote matrices and vectors, respectively. By upper case letter $\mathbf{X}, \mathbf{Y}, \mathbf{Z}$, we denote data matrices in $\mathbb{R}^{d \times T}$, where $d$ and $T$ are the dimensions of the relevant variable and the number of samples. Lower case $\mathbf{x}, \mathbf{y}, \mathbf{z}$, denote the relevant quantities of a single sample, and $\mathbf{x}_t, \mathbf{y}_t, \mathbf{z}_t$, denote the $t^{\text{th}}$ sample. $\|\mathbf{M}\|_F^2$ denotes the Frobenius norm of $\mathbf{M}$.

### 2.1 Review of predictive coding

**Probabilistic model.** In this section, we review the supervised predictive coding algorithm from [4]. The derivation starts from a probabilistic model for supervised learning, which parallels the architecture of an artificial neural network (ANN) with $n + 1$ layers. In this model, the neurons of each layer are random variables $\mathbf{z}^{(l)}$ (denoting the vector of activations in the $l^{\text{th}}$ layer) with layers 0 and $n$, respectively, denoting the input and output layers of the network. We assume that the joint probability of the latent variables factorizes in a Markovian structure

$$p(\mathbf{z}^{(0)}, \mathbf{z}^{(1)}, \cdots, \mathbf{z}^{(n)}) = p(\mathbf{z}^{(n)}|\mathbf{z}^{(n-1)}) \times p(\mathbf{z}^{(n-1)}|\mathbf{z}^{(n-2)}) \times \cdots \times p(\mathbf{z}^{(0)})$$

with the relationship between the random variables of adjacent layers given by:

$$p\big(\mathbf{z}^{(l)}|\mathbf{z}^{(l-1)}\big) = \mathcal{N}\big(\mathbf{z}^{(l)}; \boldsymbol{\mu}^{(l)}, \boldsymbol{\Sigma}^{(l)}\big), \qquad \text{with} \quad \boldsymbol{\mu}^{(l)} = \mathbf{W}^{(l-1)} f(\mathbf{z}^{(l-1)}), \quad \boldsymbol{\Sigma}^{(l)} = \sigma^{(l)2}\mathbf{I}. \quad (1)$$

The mean of the probability density on layer $l$ mirrors the activity of the analogous ANN given by $\boldsymbol{\mu}^{(l)} = \mathbf{W}^{(l-1)} f(\mathbf{z}^{(l-1)})$, where $\mathbf{W}^{(l-1)}$ are the weights connecting layers $l-1$ and $l$. The objective function is then given by the negative log-likelihood of the joint distribution function:

$$L = -\sum_t \log p(\mathbf{z}_t^{(0)}, \ldots, \mathbf{z}_t^{(n)}) = \frac{1}{2} \sum_l \frac{\big\|\mathbf{Z}^{(l)} - \mathbf{W}^{(l-1)} f(\mathbf{Z}^{(l-1)})\big\|_F^2}{\sigma^{(l)2}} + \text{const}, \qquad (2)$$

where we have switched to the data matrix notation for brevity and assumed the variances $\sigma^{(l)2}$ are fixed hyperparameters. In the following, we refer to $L$, Eq. (2), without the constant term.

**Learning.** Learning in the Whittington and Bogacz framework takes place in two steps [4]. First, the values of the random variables are determined by finding the most probable configuration of the joint distribution function when both the input and output layers are conditioned on the given input and output ($\mathbf{z}^{(0)} = \mathbf{x}$ and $\mathbf{z}^{(n)} = \mathbf{y}$):

$$\mathbf{z}^{*(1)}, \ldots, \mathbf{z}^{*(n-1)} = \underset{\mathbf{z}^{(1)},\ldots,\mathbf{z}^{(n-1)}}{\arg \min} \ L(\mathbf{z}^{(0)} = \mathbf{x}, \mathbf{z}^{(n)} = \mathbf{y}). \tag{3}$$

The solution to this minimization problem can be found via gradient descent, which we evaluate component-wise for clarity as:

$$\dot{z}_j^{(l)} = -\eta \partial_{z_j} L = \eta(-\varepsilon_j^{(l)} + \sum_i \varepsilon_i^{(l+1)} W_{ij}^{(l)} f'(z_j^{(l)})), \quad \text{with} \quad \varepsilon_i^{(l)} = \frac{z_i^{(l)} - \mu_i^{(l)}}{\sigma^{(l)2}}, \tag{4}$$

where $\eta$ is the gradient descent step size. The second step is then to minimize the objective with respect to the weights while keeping the previously obtained neuron values fixed. This corresponds to optimizing the value of the loss at the MAP estimate and can also be carried out by gradient descent, leading to:

$$\delta W_{ij}^{(l)} \propto -\frac{\partial L}{\partial W_{ij}^{(l)}} = \varepsilon_i^{*(l+1)} f(z_j^{*(l)}), \tag{5}$$

where the star superscript denotes equilibrium values of the latent variable dynamics obtained in the first step (Eq. (3)). This algorithm can be implemented by a biologically plausible network as described in [4]; see Figure 1a. However, as discussed in Section 1, its mapping onto the cortex has proved controversial. For further details regarding these steps see [4].

## 3 A constrained predictive coding framework

In this section, we introduce and discuss our novel covariance-constrained predictive coding (CCPC) model that builds upon the supervised PC paradigm of [4]. Our model also straightforwardly extends to the unsupervised PC framework of [1, 2]. For simplicity, we work in the linear regime ($f(x) = x$), which allows us to prove different properties of our framework.

### 3.1 Derivation of an upper bound objective

**Reduction to a sum of objectives.** We start by reducing the optimization problem, Eq. (2), into a set of overlapping sub-problems, which will allow us to break the symmetry between feedforward and feedback weights. To do so, we first introduce a copy of the terms containing the weights $\mathbf{W}^{(1)}$ to $\mathbf{W}^{(n-2)}$, denoted by $\mathbf{W}_a^{(l)}$ and $\mathbf{W}_b^{(l)}$ respectively, as

$$\min_{\mathbf{Z},\mathbf{W}} L = \min_{\mathbf{Z},\mathbf{W}} \frac{1}{2} \sum_{l=1}^n \frac{\left\|\mathbf{Z}^{(l)} - \mathbf{W}^{(l-1)}\mathbf{Z}^{(l-1)}\right\|_F^2}{\sigma^{(l)2}} = \min_{\mathbf{Z},\mathbf{W}_a,\mathbf{W}_b} \hat{L}$$

$$\hat{L} = \min_{\mathbf{Z},\mathbf{W}_a,\mathbf{W}_b} \frac{1}{2} \frac{\left\|\mathbf{Z}^{(1)} - \mathbf{W}_b^{(0)}\mathbf{Z}^{(0)}\right\|_F^2}{\sigma^{(1)2}} + \frac{1}{2} \frac{\left\|\mathbf{Z}^{(n)} - \mathbf{W}_a^{(n-1)}\mathbf{Z}^{(n-1)}\right\|_F^2}{\sigma^{(n)2}} \tag{6}$$

$$+ \frac{1}{4} \sum_{l=2}^{n-1} \left[ \frac{\left\|\mathbf{Z}^{(l)} - \mathbf{W}_b^{(l-1)}\mathbf{Z}^{(l-1)})\right\|_F^2}{\sigma^{(l)2}} + \frac{\left\|\mathbf{Z}^{(l)} - \mathbf{W}_a^{(l-1)}\mathbf{Z}^{(l-1)})\right\|_F^2}{\sigma^{(l)2}} \right].$$

For consistency, we rename $\mathbf{W}^{(0)}$, $\mathbf{W}^{(n-1)}$ to $\mathbf{W}_b^{(0)}$, $\mathbf{W}_a^{(n-1)}$, respectively. Introducing these copies does not change the optimization[2] but will help us avoid weight sharing in the steps below. We now pair the terms two by two as

$$\min_{\mathbf{Z},\mathbf{W}_a,\mathbf{W}_b} \hat{L} = \frac{1}{2} \sum_{l=1}^{n-1} \left[ g_b^{(l)} \left\|\mathbf{Z}^{(l)} - \mathbf{W}_b^{(l-1)}\mathbf{Z}^{(l-1)}\right\|_F^2 + g_a^{(l)} \left\|\mathbf{Z}^{(l+1)} - \mathbf{W}_a^{(l)}\mathbf{Z}^{(l)}\right\|_F^2 \right], \tag{7}$$

---

[2]This can be directly verified by finding the optima for $\mathbf{W}$'s before and after the change. We have $\mathbf{W}^{(l)} = \mathbf{W}_a^{(l)} = \mathbf{W}_b^{(l)} = \mathbf{Z}^{(l+1)}\mathbf{Z}^{(l)\top}(\mathbf{Z}^{(l+1)}\mathbf{Z}^{(l)\top})^{-1}$. Plugging these back into Eq. (6) we see that the equality holds. However, in the next step, since we treat $\mathbf{W}_a$'s and $\mathbf{W}_b$'s differently, $\mathbf{W}_a = \mathbf{W}_B$ will no longer hold.

with $g_a^{(n-1)} = 1/\sigma^{(n)2}$, $g_b^{(1)} = 1/\sigma^{(1)2}$, and $g_a^{(l-1)} = g_b^{(l)} = 1/(2\sigma^{(l)2})$ for $l = 2, \ldots, n-1$.

Weight sharing occurs here from terms like $\mathbf{z}^{(l+1)\top}\mathbf{W}\mathbf{z}^{(l)}$, obtained from expanding the squared norms in Eq. (7). Indeed, the gradient descent dynamics with respect to $\mathbf{z}^{(l+1)}$ (resp. $\mathbf{z}^{(l)}$) leads to terms of the form $\mathbf{W}\mathbf{z}^{(l)}$ in $\dot{\mathbf{z}}^{(l+1)}$ (resp. $\mathbf{W}^\top\mathbf{z}^{(l+1)}$ in $\dot{\mathbf{z}}^{(l)}$), which use the same weights $\mathbf{W}$. Thanks to the doubling of the weights, we can avoid this problem by optimizing each term in the sum in Eq. (7) separately. In other words,

$$\min_{\mathbf{Z},\mathbf{W}_a,\mathbf{W}_b} \hat{L} \leq \sum_{l=1}^{n-1} \min_{\mathbf{Z}^{(l)},\mathbf{W}_a^{(l)},\mathbf{W}_b^{(l-1)}} \hat{L}^{(l)} , \tag{8}$$

$$\text{where } \hat{L}^{(l)} \equiv \frac{1}{2}\sum_{l=0}^{n}\left[g_b^{(l)}\left\|\mathbf{Z}^{(l)} - \mathbf{W}_b^{(l-1)}\mathbf{Z}^{(l-1)}\right\|_F^2 + g_a^{(l)}\left\|\mathbf{Z}^{(l+1)} - \mathbf{W}_a^{(l)}\mathbf{Z}^{(l)}\right\|_F^2\right] .$$

This inequality holds simply because we are no longer finding the minimum of the full objective $\hat{L}$. We are instead finding the minimum of each component separately and then evaluating $\sum_l \hat{L}^{(l)} = \hat{L}$. This splits the $(n+1)$-layer optimization problem into a set of 3-layer optimizations, in each of which only the middle layer is being optimized. Note, however, that these are overlapping, so the different optimization problems need to be solved self-consistently. We make this precise in the supplementary materials section and show that it provides an upper bound for our objective $L$ (SM Sec. A). Separating the objective function in this manner eliminates the weight sharing problem for $\mathbf{W}_b$, but the problem remains for $\mathbf{W}_a$. This can be seen by noting that the gradient descent equations only depend on $\mathbf{W}_b$ but depend on both $\mathbf{W}_a$ and $\mathbf{W}_a^\top$. We address this problem next.

**Whitening constraint.** The idea of decorrelating internal representation has been widely used for unsupervised tasks, often motivated by neuroscience [36–38]. In the case of deep learning, the main motivations were improved convergence speed and generalization [39–41]. Decorrelation has also been used to circumvent the weight transport problem [55]. Inspired by these observations, we introduce the inequality constraint $\frac{1}{T}\mathbf{Z}^{(l)}\mathbf{Z}^{(l)\top} \preceq \mathbf{I}$, imposing an upper bound on the eigenvalues of the covariance matrix. In Sec. 5 and SM Sec. B we show that under some assumptions this inequality constraint leads to whitening the latent variables. Adding this inequality constraint to the objective we have:

$$\min_{\mathbf{Z}^{(l)},\mathbf{W}_a^{(l)},\mathbf{W}_b^{(l)}} \hat{L}^{(l)} \leq \min_{\substack{\mathbf{Z}^{(l)},\mathbf{W}_a^{(l)},\mathbf{W}_b^{(l)} \\ \mathbf{Z}^{(l)}\mathbf{Z}^{(l)\top}\preceq T\times\mathbf{I}}} \hat{L}^{(l)}$$

$$\leq \min_{\substack{\mathbf{Z}^{(l)},\mathbf{W}_a^{(l)},\mathbf{W}_b^{(l)} \\ \mathbf{Z}^{(l)}\mathbf{Z}^{(l)\top}\preceq T\times\mathbf{I}}} \frac{1}{2}\left[g_b^{(l)}\left\|\mathbf{Z}^{(l)} - \mathbf{W}_b^{(l-1)}\mathbf{Z}^{(l-1)}\right\|_F^2 + c^{(l)}\left\|\mathbf{Z}^{(l)}\right\|_F^2\right.$$

$$\left. + g_a^{(l)}\operatorname{Tr}\left(-2\mathbf{Z}^{(l+1)\top}\mathbf{W}_a^{(l)}\mathbf{Z}^{(l)}\right) + g_a^{(l)}T\,\mathbf{W}_a^{(l)\top}\mathbf{W}_a^{(l)}\right]$$

The first inequality results from the fact that the parameters of the new optimization problem are a subset of the parameters of the original optimization problem. In the second inequality we have expanded the second term of $\hat{L}^{(l)}$ and used the constraint $\frac{1}{T}\mathbf{Z}^{(l)}\mathbf{Z}^{(l)\top} \preceq \mathbf{I}$ to replace $\mathbf{Z}^{(l)}\mathbf{Z}^{(l)\top}$ with $T$, making the term larger, resulting again in an upper bound. This replacement eliminates the weight sharing problem in $\mathbf{W}_a$, as the gradient descent equations now only depend on $\mathbf{W}_a^\top$ and not on $\mathbf{W}_a$ (cf. Eqs. (10) and (11)).

The inequality can be implemented by using a positive-definite Lagrange multiplier $\mathbf{Q}^{(l)\top}\mathbf{Q}^{(l)}$ [67]:

$$\min_{\mathbf{Z}^{(l)},\mathbf{W}_a^{(l)},\mathbf{W}_b^{(l-1)}} \max_{\mathbf{Q}^{(l)}} \frac{1}{2}\left[g_b^{(l)}\left\|\mathbf{Z}^{(l)} - \mathbf{W}_b^{(l-1)}\mathbf{Z}^{(l-1)}\right\|_F^2 + c^{(l)}\left\|\mathbf{Z}^{(l)}\right\|_F^2\right.$$

$$+ g_a^{(l)}\operatorname{Tr}\left(-2\mathbf{Z}^{(l+1)\top}\mathbf{W}_a^{(l)}\mathbf{Z}^{(l)}\right) + g_a^{(l)}T\,\mathbf{W}_a^{(l)\top}\mathbf{W}_a^{(l)}$$

$$\left. + \operatorname{Tr}\mathbf{Q}^{(l)\top}\mathbf{Q}^{(l)}\left(\mathbf{Z}^{(l)}\mathbf{Z}^{(l)\top} - T\mathbf{I}\right) + \text{const}\right], \tag{9}$$

where we have added an additional quadratic term in $\mathbf{Z}$ as a regularizer. This can also be interpreted as an extra term in the prior of the latent space. Equation (9) is the final form of the objective, which we then optimize using gradient descent/ascent.

## 3.2   Neural dynamics and learning rules.

Similar to the PCN of [4], the dynamics of our network during learning proceeds in two steps. First the neural dynamics is derived by taking gradient steps of $\hat{L}^{(l)}$ from Eq. (9) with respect to $\mathbf{z}^{(l)}$:

$$\dot{\mathbf{z}}^{(l)} = g_b^{(l)} \mathbf{W}_b^{(l-1)} \mathbf{z}^{(l-1)} + g_a^{(l)} \mathbf{W}_a^{(l)\top} \mathbf{z}^{(l+1)} - \big(g_b^{(l)} + c^{(l)}\big) \mathbf{z}^{(l)} - g_a^{(l)} \mathbf{Q}^{(l)\top} \mathbf{n}^{(l)}, \qquad (10)$$

where we have defined the variables $\mathbf{n}^{(l)} = (1/g_a^{(l)}) \mathbf{Q}^{(l)} \mathbf{z}^{(l)}$ for each layer of the network.

The weight updates are derived via stochastic gradient descent of the loss given in Eq. (9) after the neural dynamics have reached equilibrium. These are given by

$$\delta \mathbf{W}_b^{(l)} \propto \Big[ g_a^{(l+1)} \big( \mathbf{W}_a^{(l+1)\top} \mathbf{z}^{(l+2)} - \mathbf{Q}^{(l+1)\top} \mathbf{n}^{(l+1)} \big) - c^{(l+1)} \mathbf{z}^{(l+1)} \Big] \mathbf{z}^{(l)\top}, \qquad (11a)$$

$$\delta \mathbf{W}_a^{(l)\top} \propto \mathbf{z}^{(l)} \mathbf{z}^{(l+1)\top} - \mathbf{W}_a^{(l)\top}, \qquad (11b)$$

$$\delta \mathbf{Q}^{(l)} \propto \mathbf{n}^{(l)} \mathbf{z}^{(l)\top} - \mathbf{Q}^{(l)}. \qquad (11c)$$

We used the neural dynamics equilibrium equation for $\mathbf{z}^{(l)}$ to simplify the weight update for $\mathbf{W}_b^{(l)}$. This yields our online algorithm (Alg. 1 in SM Sec. E), with the architecture shown in Fig. 1b. The algorithm can be implemented in a biologically plausible neural network as in Fig. 2; see Sec. 4.

## 4   Biological implementation and comparison with experimental observations

In this section, we introduce a biologically plausible neural circuit that implements the CCPC algorithm, denoted by BioCCPC. We also demonstrate that the details of this circuit resemble the neurophysiological properties of pyramidal cells in the neocortex and the hippocampus.

**Neural architecture.**   The algorithm for BioCCPC (Alg. 1) summarized by the neural dynamics from Eq. (10) and weight update rules from Eqs. (11) can be implemented by a neural circuit with schematic shown in Fig. 2.

The activity of the $n-1$ hidden layers, $\{\mathbf{z}^{(l)}\}_{l=1}^{n-1}$, is encoded as the outputs of $n-1$ sets of neurons, representing pyramidal neurons of different cortical regions. The matrices $\mathbf{W}_a^{(l)}$ and $\mathbf{W}_b^{(l)}$ are encoded as the weights of synapses between the pyramidal neurons of adjacent layers. Explicitly, the matrix $\mathbf{W}_a^{(l)}$ (resp. $\mathbf{W}_b^{(l-1)}$) is the efficacy of the synapse connecting $\mathbf{z}^{(l+1)}$ (resp. $\mathbf{z}^{(l-1)}$) to the pyramidal neurons $\mathbf{z}^{(l)}$. Because of the disjoint nature of these two inputs, we model these as synapsing respectively onto the distal (apical tuft) and proximal (mostly basal) dendrites of the pyramidal neurons, respectively; see Fig. 2. This is reminiscent of cortical pyramidal neurons, which also have two integration sites, the proximal compartment comprised of the basal and proximal apical dendrites providing inputs to the soma, and the distal compartment comprised of the apical dendritic tuft [22, 68]. These two compartments receive excitatory inputs from two separate sources [18, 69].

Similarly, the auxiliary variables $\mathbf{n}^{(l)}$ are represented by the activity of interneurons in each cortical region.[3] The $\mathbf{Q}^{(l)}$ synaptic weights are encoded in the weights of synapses connecting $\mathbf{n}^{(l)}$ to $\mathbf{z}^{(l)}$, while $\mathbf{Q}^{(l)\top}$ models the weights of synapses from $\mathbf{z}^{(l)}$ to $\mathbf{n}^{(l)}$. In a biological setting, the implied equality of weights of synapses from $\mathbf{z}^{(l)}$ to $\mathbf{n}^{(l)}$ and the transpose of those from $\mathbf{n}^{(l)}$ to $\mathbf{z}^{(l)}$ can be guaranteed by applying the same Hebbian learning rule [55]. However, unlike previous work, we do not require interneurons of one layer to be connected to the pyramidal neurons of another layer.

**Neural dynamics and canonical components.**   The neural dynamics of the pyramidal neurons of our circuit given by Eq. (10) can be recast as the dynamics of a three compartment neuron:

$$\tau \dot{\mathbf{z}}^{(l)} = -g_{\text{lk}}^{(l)} \mathbf{z}^{(l)} + g_a^{(l)} (\mathbf{v}_a^{(l)} - \mathbf{z}^{(l)}) + g_b^{(l)} (\mathbf{v}_b^{(l)} - \mathbf{z}^{(l)}), \qquad (12)$$

where we have defined the apical and basal compartmental membrane potentials

$$\mathbf{v}_a^{(l)} = \mathbf{W}_a^{(l)\top} \mathbf{z}^{(l+1)} - \mathbf{Q}^{(l)\top} \mathbf{n}^{(l)}, \qquad \mathbf{v}_b^{(l)} = \mathbf{W}_b^{(l-1)} \mathbf{z}^{(l-1)}. \qquad (13)$$

---

[3]There are multiple types of interneurons targeting pyramidal cells [70, 71]. The interneurons of BioCCPC most closely resemble the somatostatin-expressing interneurons, which preferentially inhibit the apical dendrites.

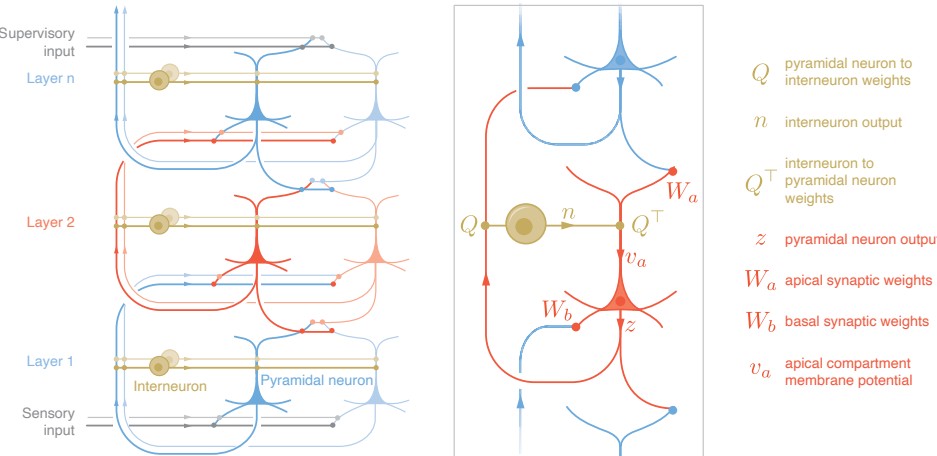

**Figure 2:** Schematic of the biological implementation of the BioCCPC algorithm. Left: overall connectivity pattern, showing all-to-all connections between pyramidal neurons in consecutive layers, and between pyramidal and interneurons in the same layer. Right: zoom in on one neuron and its immediate neighbors, showing the membrane potentials and synaptic weights that are relevant for our algorithm.

Here we defined the leak conductance $g_{\text{lk}}^{(l)} = c^{(l)} - g_a^{(l)}$. Interestingly, this is the same neural dynamics of a three-compartment neuron posited previously in [28, 30], which we derived in a normative way. Here, $g_a$ (resp. $g_b$) are conductances between the apical (resp. basal) compartments and the soma, and $g_{\text{lk}}$ is the somatic leak. Our derivation clarifies the relationship between the physiological quantities and the objective function. Indeed, the basal and apical conductances play the role of the inverse variances in the predictive coding objective. These inverse variances, generally referred to as 'precisions' in the PC literature, have previously been argued to be encoded in lateral inhibitions [2] or top-down attention [72, 73]. In our model, they are simply encoded in the compartmental conductances. In Sec. 5 we show that the leak term also has a functional role and determines a threshold for dynamically learning more diverse variables [74, 75].

**Synaptic weight updates.** Matching observations in the cortex, in our circuit [18, 22, 68, 76, 77], the basal and apical weights ($\mathbf{W}_b$ and $\mathbf{W}_a$) of our algorithm are updated differently. The basal synaptic weights of the pyramidal neurons, given by the elements of $\mathbf{W}_b^{(l)}$, are updated by the product of two factors represented in the corresponding post- and pre-synaptic neurons (Eq. 11a):

$$\delta \mathbf{W}_b^{(l-1)} \propto \left[ g_a^{(l)} \mathbf{v}_a^{(l)} - c^{(l)} \mathbf{z}^{(l)} \right] \mathbf{z}^{(l-1)\top} . \tag{14}$$

The first term in parentheses, $g_a^{(l)} \mathbf{v}_a^{(l)}$, is proportional to the total apical current, given by the difference between the excitatory synaptic current in the apical tuft, $\mathbf{W}_a^{(l)\top} \mathbf{z}^{(l+1)}$, and the inhibitory current induced by interneurons synapsing onto the distal compartment, $\mathbf{Q}^{(l)\top} \mathbf{n}^{(l)}$. Biologically, this factor can be approximated by the calcium plateau potential traveling down the apical shaft. The calcium plateau potential has been experimentally seen to drive plasticity of the basal synapses matching the derived update rule of our circuit [18, 20, 23–25]. The second factor is simply the neural output. Because the total update is not purely dependent on the action potentials of the pre- and post-synaptic neurons, such plasticity is called non-Hebbian [25]. The synaptic learning rule for the apical weights $\mathbf{W}_a^{(l)}$ (Eq. 11b) and the synapses connecting the interneurons and the pyramidal neurons $\mathbf{Q}^{(l)}$ (Eq. 11c) are simply Hebbian, also matching experimental observations in the cortex [19, 78].

**Features and limitations.** Here we summarize the main differences between the biological implementations of CCPC and PC. In PC, the forward and backward weights (in our notation $\mathbf{W}_b$ and $\mathbf{W}_a^\top$) are symmetric, whereas, in CCPC, they are not constrained to be symmetric and are generically not so (see Sec. 6). Another difference is that in PC, the connection between value neurons and error neurons is one-to-one and fixed; this constraint does not exist between the pyramidal and interneurons of our model, which are no longer one-to-one, and have plastic connectivity.

While we have shown the improvement in biological realism of our model over the traditional PC network, this was done in the linear case. However, recent work on deep linear networks [79] has provided many insights into the learning dynamics of deep networks. Some of the properties discovered in the deep linear network, like "balancedness" of weights [80, 81], generalize for certain nonlinear networks [81]. We believe several of our observations will generalize in a similar fashion.

There are other aspects that can also be improved. For example, the connectivity between pyramidal neurons and interneurons of CCPC is required to be symmetric. This symmetry can be achieved via Hebbian learning [55]; however, it would be interesting to explore if such symmetry is indeed required. In CCPC, symmetric connectivity is only required between neurons of the *same* cortical region, which is a much less stringent biological requirement than that of symmetric weights between *different* cortical regions, as in PC. Another shortcoming of our model is that while the teaching signal for the basal synapses in CCPC is signed and graded, in the cortex these signals are generally believed to be stereotypical [18].

## 5 Theoretical arguments and interpretation

In this section, we summarize some of the salient theoretical features of our framework and describe their biological significance. Further details and proofs are given in SM Sec. B.

### 5.1 Implicit error computation

In backpropagation, as well as in PC, the learning algorithm computes a loss (or a local error) which is then used to compute updates to the weights. In CCPC, no error or loss is explicitly computed. So how does CCPC learn? In the following proposition, we show that in the $g_a \to 0$ limit, the algorithm implicitly computes an error in its weight updates. This quasi feed-forward or weak nudging limit has been used to explore the learning dynamics of biologically plausible networks [30] and is related to the 'fixed prediction' assumption [4, 61, 62].

**Proposition 1.** *Assume that the learning rules (Eqs. 11) are at equilibrium and we receive a new datapoint given by* $\mathbf{x}_{T+1}, \mathbf{y}_{T+1}$*. For* $c^{(l)} = 0$ *and in the limit of* $\epsilon \equiv g_a/g_b \to 0$*, the leading term in* $\epsilon$ *for the forward weight updates* $\delta\mathbf{W}_b^{(l-1)}$ *for this new sample is given by*

$$\delta\mathbf{W}_b^{(l-1)} \propto \mathbf{v}_{a,T+1}^{(l)} \mathbf{z}_{T+1}^{(l-1)\top} = \epsilon^{n-l} \left[ \mathbf{W}_a^{(l)\top} \cdots \mathbf{W}_a^{(n-1)\top}(\mathbf{y}_{T+1} - \tilde{\mathbf{y}}_{T+1}) \right] \mathbf{z}_{T+1}^{(l-1)\top} + \mathcal{O}(\epsilon^{n-l+1}),$$

*where* $\tilde{\mathbf{y}}_{T+1} = \mathbf{Y}\mathbf{Z}^{(l-1)\top} \left( \mathbf{Z}^{(l-1)}\mathbf{Z}^{(l-1)\top} \right)^{-1} \mathbf{z}_{T+1}^{(l-1)}$ *is the optimal linear inferred value for* $\mathbf{y}_{T+1}$*.*

**Biological interpretation**  Recall that the learning rule for the basal synapses is proportional to $\mathbf{v}_a^{(l)}$ (Eq. (14)). We argued in the previous section that this quantity can be implemented as the calcium plateau potential. In this light, Prop. 1 shows that the calcium plateau potential is implicitly encoding the prediction error. This is remarkable as in our algorithm, neither the prediction nor the prediction error is explicitly computed. However, via the combined learning dynamics of the synapses and the neural dynamics of the interneurons, the final apical signal implicitly encodes the prediction error. This is in spirit similar to difference target propagation (DTP) [47], however, in DTP, the difference between the forward pass and the target is *explicitly* computed and backpropagated.

### 5.2 Adaptive latent dimension

For optimization within a single hidden layer, we show that the system performs adaptive latent dimension discovery by thresholding eigenvalues of a Gramian including contributions from both adjacent layers. If we take the loss function in (9), setting $g_a = 1, g_b = \epsilon$, and optimizing over $\mathbf{W}_a, \mathbf{W}_b$, we get the objective $\text{Tr}[(c+\epsilon)\mathbf{Z}^\top\mathbf{Z} - \mathbf{Z}^\top\mathbf{Z}[\epsilon\mathbf{X}^\top(\mathbf{X}\mathbf{X}^\top)^{-1}\mathbf{X} + \mathbf{Y}^\top\mathbf{Y}]$ for $\mathbf{Z}$. The next theorem indicates the solution:

**Theorem 1.** *Let the concatenated matrix* $\boldsymbol{\Xi} = [\epsilon^{1/2}\mathbf{C}_x^{-1/2}\mathbf{X}, \mathbf{Y}] \in \mathbb{R}^{(d_X+d_Y)\times T}$ *have SVD* $\boldsymbol{\Xi} = \sum_{\alpha=1}^{(d_X+d_Y)} \mathbf{u}_\alpha \lambda_\alpha \mathbf{v}_\alpha^\top$ *with* $\{\mathbf{u}_\alpha \in \mathbb{R}^{(d_X+d_Y)}\}$ *and* $\{\mathbf{v}_\alpha \in \mathbb{R}^T\}$ *both being sets of orthonormal vectors, with the convention that* $(\lambda_\alpha)$ *are sorted in decreasing order. We consider the minimization*

$$\min_{\mathbf{Z}\in\mathbb{R}^{d\times T}, \mathbf{Z}\mathbf{Z}^\top \preccurlyeq I_d} \text{Tr}\left[(c+\epsilon)\mathbf{Z}^\top\mathbf{Z} - \mathbf{Z}^\top\mathbf{Z}[\epsilon\mathbf{X}^\top(\mathbf{X}\mathbf{X}^\top)^{-1}\mathbf{X} + \mathbf{Y}^\top\mathbf{Y}]\right]. \tag{15}$$

*Then one of the optimal solutions is given by $\hat{\mathbf{Z}} = \sum_{\alpha=1}^{D} \mathbf{w}_\alpha \mathbb{1}(\lambda_\alpha > c + \epsilon)\mathbf{v}_\alpha^\top$ where $\{\mathbf{w}_\alpha \in \mathbb{R}^d\}$ is an arbitrary set of orthonormal vectors, with $D = \min(d, d_X + d_Y)$.*

Thus, the latent modes are found by thresholding the eigenvalues [75] of the sum of Gramians [74] in Eq. (15). From this theorem, we derive the computational limits of $\epsilon \to \infty$ and $\epsilon \to 0$ in the supplementary materials. We also show that the neurons of the narrowest layer get whitened and the other layers become low-rank. For further details of these statements and all proofs see SM Sec. B.

**Biological interpretation** In our model, the number of neurons in the intermediate layers is fixed. However, the dimensionality of the representation encoded in these intermediate layers is determined dynamically according to the spectrum of the eigenvalue problem specified in Thm. 1. In particular, it is the quantity $c^{(l)}$, which encodes the somatic compartment leak conductance, that biologically sets an adaptive threshold for the dimensionality of the latent representation. In other words, lower (resp. higher) somatic leak corresponds to higher (resp. lower) sensitivity to the small eigenvalues of the above eigenvalue problem. For details of the interpretation of the eigenvalue problem in different limits ($\epsilon \to \infty$ and $\epsilon \to 0$), see SM Sec. B.

## 6   Numerical experiments

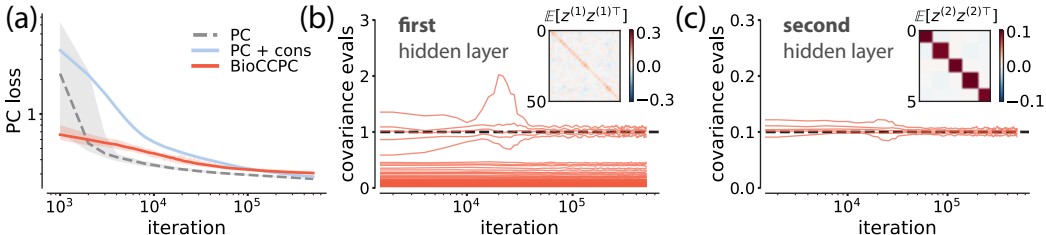

**Figure 3:** Our algorithm (BioCCPC) compares well to less biologically plausible predictive-coding (PC) implementations while enforcing an inequality constraint on the covariance of hidden-layer activations. Simulations run on the MNIST dataset using a network with two hidden layers, sizes 50 and 5. The shaded area represents the 95% range out of 40 runs. (a) Evolution of the predictive-coding loss, Eq. (6), on a validation set during learning. (b), (c) Evolution of the eigenvalues of the covariance matrix of hidden-layer activations in the two hidden layers. The dashed lines show the scale of the constraint; see SM Sec. A. Inset: covariance matrix at the end of training. Note how the activity is whitened in the second hidden layer (c), and low-rank in the first hidden layer (b).

We tested our BioCCPC algorithm on the MNIST [82] as well as other datasets.[4] We compared our results with a standard predictive coding network [4] with linear activation (PC) and with a version that includes a covariance constraint (PC+cons; Sec. C). We used unit variances, $\sigma^{(l)2} = 1$, in the PC networks, and matching conductances in our algorithm, $g_b^{(1)} = g_a^{(n)} = 1$, $g_a^{(l)} = g^{(l)} = \frac{1}{2}$ for other $l$.

Our method finds solutions that are almost optimal in terms of predictive-coding loss on a held-out validation set (see Fig. 3a) and are generally better than the solutions obtained from simply adding the covariance constraint to a predictive-coding network. Learning is slower in our network than in unconstrained PCN, as it takes time for the covariance constraint to be enforced. It is possible that different initialization schemes could improve the learning speed, but we leave this for future work. Note that since we focus on linear networks, metrics based on prediction accuracy are not very informative for the datasets we consider, which are not linearly separable.

The covariance constraint is saturated in the narrowest layer—in our example, the second hidden layer, of size 5—see Fig. 3c. The activation in the other hidden layer is rank-restricted by the narrowest layer, as described above, and so it is only whitened within a 5D subspace, Fig. 3b. In SM Sec. C, we provide further numerical experiments, exploring the effects of thresholding, the size of the network, and performance on other datasets. In all tested scenarios, BioCCPC achieves the same validation loss as PC (Figs. 5 and 6 of SM Sec. C). We also show that the forward and backward weights, $\mathbf{W}_b$ and $\mathbf{W}_a^\top$, are not symmetric, in accordance with biological observations, and explore their relationship.

---

[4]The code is available at `https://github.com/ttesileanu/bio-pcn`.

In SM Sec. D we compare PC and BioCCPC with regards to generalization. We find that there is very little generalization gap for either case (Fig. 9). This is due to the linearity of our model, which acts as a strong regularizer. Because of this, the potential generalization benefits of whitening the latent variables are not observed. Therefore, the primary benefit of the covariance constraint in this work is the elimination of weight sharing in $\mathbf{W}_a$ (cf. Sec. 3.1).

## 7 Conclusion

We have derived a biologically plausible algorithm for covariance-constrained predictive coding and have shown that it avoids the criticisms of PC and has many features in common with recent experimental observations. Our algorithm does not have symmetric forward and backward connectivity matrices, and does not need one-to-one connectivity between any neurons. Furthermore, we showed that the learning rules of our algorithm closely match experimental observations, and we connected our parameters to neurophysiological quantities. Using the simplicity afforded to us because of the linearity, we showed a number of interesting properties of our algorithm, including the effect of the somatic leak term, which acts as an adaptive thresholding mechanism. We also showed how our algorithm implicitly computes a difference term that it uses for learning. We argued that this term could be encoded in the calcium plateau potential, which is known to affect the learning of basal synapses. We hope that this concrete connection between our normative model and observation can further our understanding of cortical computation by guiding future experiments.

### Acknowledgments and Disclosure of Funding

We would like to thank Colin Bredenberg for guidance and helpful comments throughout the project. SG, TT, YB, and DC were internally funded by the Simons Foundation. AS was partly funded by Simons Foundation Neuroscience grant SF 626323 during this work.

### Broader impact

We do not foresee any potential societal harm caused by our work. We hope our contribution moves forward the discussion on the relationship between multi-layer networks and cortical structure.

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
