# OpenReview forum: "Constrained Predictive Coding as a Biologically Plausible Model of the Cortical Hierarchy"
_NeurIPS.cc/2022/Conference — NeurIPS 2022 Accept_

### Official Review · Reviewer_9PPr · 2022-06-29

**Rating:** 4
**Confidence:** 4
**Soundness:** 2 fair
**Presentation:** 3 good
**Contribution:** 2 fair

**Summary:**

This paper derives an upper bound for the PC objective by imposing a decorrelation-inspired inequality constraint on the latent space, which later comes to a biologically plausible algorithm BioCCPC. The paper shows that the proposed algorithm matches the known physiology of the cortical hierarchy.


**Questions:**

Major:
Please refer to the comments above in the Weakness section for the major concerns.

Minor:
The related work section could be added more theoretical or application works that include predictive coding. Simply listing a review paper may not be a good option.


**Limitations:**

See comments above.


**Strengths And Weaknesses:**

Strengths:

(1) The paper is well written, the structure and presentation are good and clear.

(2) The idea and theoretical analysis seem sound.

Weakness:

(1) After reading [4,1,2], the level of novelty contribution is not very clear, especially in the current linear setting.

(2)  I would like to see if the authors could provide additional experimental results on more (complicated) data, and the comparison between different PC baselines.

(3) I would also suggest that the authors can provide a way to interpret the results, to show a better way to understand the model.

---

> ### Author Response · Authors · 2022-08-02
> **Response to reviewer 9PPr**
>
> We thank the reviewer for their feedback and constructive suggestions. We are pleased that the reviewer finds the idea behind our approach as well as our theoretical analysis sound. We have made some changes and additions based on the feedback outlined below.
>
> > * I would like to see if the authors could provide additional experimental results on more (complicated) data.
>
> In the submission, we tested the algorithm on MNIST and, in the supplementary materials, provided results for MediaMill (a multi-view dataset of video frame/annotation pairs). We have now expanded our empirical verification section to include a number of new experiments. In order to show that our biologically realistic algorithm performs as well as the previous non-realistic algorithm, we performed experiments on a number of other datasets: Fashion MNIST (FMNIST), CIFAR10, CIFAR100, and Labeled Faced in the Wild (LFW). The FMNIST and CIFAR experiments were done in the supervised setting. The experiments on the MediaMill and LFW dataset (which comprises two views of different faces) were performed in a self-supervised fashion with an auto-encoder structure (i.e. using one view as the input $\mathbf x$ and the other as output $\mathbf y$).
>
> In all tested scenarios (MNIST, FMNIST, CIFAR10, CIFAR100, MediaMill, LFW), Bio-PCN achieves the same loss as PCN, demonstrating that optimizing the upper bound that we derived effectively optimizes the original objective.  The new experimental results can be found in the updated manuscript  (supplementary materials Sec. C.2).
>
> > * Comparison between different PC baselines.
>
> The PC implementation of Whittington and Bogacz that we compare against is a standard implementation. We are not aware of any other PC baseline that performs differently compared to this. We would be happy to provide the comparison if the reviewer can point us in the right direction.
>
> >* (Level of novelty) After reading [4,1,2], the level of novelty contribution is not very clear, especially in the current linear setting.
> >
> The primary focus of the work is to show that the predictive coding framework has an implementation which is not only biologically realistic, but can also shed light on a number of recent experimental observations (e.g. the calcium plateau potential being the teaching signal is explained by Prop. 1). We believe this is significant since the known biology of the cortex has lead researchers to doubt the validity of the predictive coding framework as a viable framework for computations in the brain (e.g. see [1]). We have also shown a number of theoretical results that we believe would be of interest to the community, both in neuroscience and in the field of deep linear networks.
>
> >* I would also suggest that the authors can provide a way to interpret the results, to show a better way to understand the model.
>
> This is a great suggestion. We agree that our theoretical results and description of the network and relation to biology can be made more clear. Our theoretical results, in terms of learning (prop 1 describes how the error is implicitly computed), whitening (theorem 1 shows that the latent variables are whitened with some assumptions), and the rank structure of the latents (theoretical results in section B.3), all have intuitive interpretations and explanations. For example, the whitening of the latent variables is implemented by the interneurons which are known to promote diversity. We will use the additional page partly to add further explanations and appropriate references to further clarify these points.
>
> >* (Related Works section) The related work section could be added more theoretical or application works that include predictive coding. Simply listing a review paper may not be a good option.
>
> We omitted this because of space constraints. But we agree with the reviewer that this is a good idea and will include a 'Related Works' section as part of the extra page provided to us if we get accepted.
> ___
> [1] Micha Heilbron and Maria Chait. Great Expectations: Is there Evidence for Predictive Coding in Auditory Cortex? Neuroscience, 389:54–73, 2018.

---

> > ### Comment · Reviewer_9PPr · 2022-08-08
> > **Thanks for the response**
> >
> > First, I appreciate the authors provided detailed responses. The new experimental results look promising, but adding them as additional support for the paper will introduce too many new materials compared with the original version. So I think another round of review will be beneficial. Besides, even though the page is limited, some of the important discussions should not be omitted.
> >
> > An important thing that I need to bring to the AC and other reviewers' attention is that, the current submission is 10-page, while the NeurIPS policy is max 9-page. "Submissions are limited to nine content pages, including all figures and tables; additional pages containing the NeurIPS paper checklist and references are allowed.  Submissions that violate the NeurIPS style (e.g., by decreasing margins or font sizes) or page limits may be rejected without further review. Papers may be rejected without consideration of their merits if they fail to meet the submission requirements, as described in this document. "

---

> > > ### Author Response · Authors · 2022-08-08
> > > **Response to reviewer 9PPr**
> > >
> > > * We first want to thank the reviewer for pointing out that we are over 9 pages. Adding the extra explanations pushed us over the page limit but we have now corrected the issue.
> > >
> > > * On the new additions: we would like to point out that the new experimental results are just further verification of our algorithm on 'more (complicated) data' as requested by the reviewer. We have not changed the main thrust or the arguments of the paper, just added further verifications and extra details as requested. (Even the setup of the experiments are not so different from what was already in the submission. The new FMNIST, CIFAR10, CIFAR100 experiments are similar to the previous MNIST experiment. The new LFW experiment is similar to the previous MediaMill experiment. The main differences are the choice of datasets which are now more complex.)  The goal of these new experiments is to provide further numerical evidence in addition to the theoretical arguments already in our original submission.

---

### Official Review · Reviewer_9q1E · 2022-07-06

**Rating:** 6
**Confidence:** 4
**Soundness:** 2 fair
**Presentation:** 3 good
**Contribution:** 2 fair

**Summary:**

The main objective of this article is to propose a theoretical extension of the Predictive Coding (PC) algorithm, called the Covariance Constrained Predictive Coding (CCPC), to better ground PC in neurophysiology. The 2 main theoretical novelties compared to PC are: 1) the weights are separated to break symmetry between the feedforward and the feedback weights 2) the eigenvalues of the covariance matrix are upper-bonded using a Lagrange multiplier decomposition. These theoretical changes allow the authors to re-interpret CCPC in light of physiology.

**Questions:**

* The authors decided to derive the PC formulation from the probabilistic interpretation (which is good, on my opinion). But the probabilistic description of the model is incomplete as the authors do not define the factorization of the joint distribution. This lack of rigor leads to a misunderstanding : the authors are trying to minimize the negative log likelihood of the joint distribution (as this is done in Rao and Ballard [1]) as in a classical generative model, but here the authors seem to tackle another problem which is inferring the label knowing the image (i.e.  $p(z^n | z^0)$). Given Eq. 2, I imagine the considered factorization is the following one : $p(z^0, z^1, … z^n) = p(z^n | z^{n-1}) * p(z^{n-1} | z^{n-2}) … * p(z^0)$. And $p(z^0)$ is the probability of the input distribution (which is hard to assess). Actually the log probability the authors are minimizing in the right term of Eq 2 (which is also 100% fine) is log $p(z^n | z^{n-1}, z^{n-2}, ..,  z^{0})$, which is equivalent (provided markovian hypothesis, which seems to be the case here) to $p(z^n | z^0)$. I would suggest to update the model section of the article to make it more accurate (just keep in mind the reader does not have to guess the authors hypothesis !).

* In general, I think the section 3.1 is the more important one (this is where the novelty of this paper is). But this part is not rigorous enough for 2 reasons : 1- The authors suggest that splitting $W$ into $W_a$ and $W_b$, doesn’t change the optimization problem. This specific point should be demonstrated. The gradient (w.r.t. $z^l$) depends now on $W_a$ and $W_b$. Meaning that if $W_a \neq W_b$ (I guess this is the case, otherwise I don’t see the point of splitting), the gradient has therefore a different value, which suppose that the optimization problem is not the same anymore. I think this part is crucial, it would deserve a more rigorous demonstration. 2 - I have trouble to understand the inequality presented in Eq. 8. In the right term of Eq 8 there is a minimization w.r.t $W_{b}^{(l)}$, but as expected just after this equation, $L^{(l)}$ does not depend on $W_{b}^{(l)}$ but rather on $W_{b}^{(l-1)}$. Is it a typo in your inequality in Eq.8 ? Even if it is a typo, this inequality would deserve more explanation.

* In Eq.9, the authors call the term $||z^{(l)}||^2$ a ‘prior’.‘Prior’ has a probabilist connotation, and here it doesn’t fit into your probabilist description of the model (nowhere, you have exhibited a normal prior for every $z^{(i)}$ in section 2.1). You can easily circumvent that by calling the term $||z^{(l)}||^2$ a ‘regularization term’ which has no probabilist connotation (I know I am being picky, but as this article is almost fully theoretical, it should be as rigorous as possible)

* Section 6: I do not understand the authors experimental comparison. It seems that the authors are comparing the training loss of PC, PC + cons, BioCCPC (cause on my understanding you have to give both the input and the one-hot to the model). Is it True ? Purely in terms of (training) loss, it seems that the BioCCPC performs not as good as PC (see Fig 3). Then what's the point of BioCCPC ? It is only a more bio-plausible formulation (if yes, it should be fitted to physiological data, see the last bullet point).

* Section 6 again: I do not think the training loss is a good measure to compare algorithms. As the authors already mention (line 141), the whitening regularization added in the CCPC should improve the generalization abilities. But with the training loss only, how is it possible to assess the generalization gap (different between training and testing ? ). Why don’t you show the testing accuracy / testing loss ?

* The authors suggest that their CCPC allows to interpret the variable in terms of neurophysiology. Then I would suggest to conduct quantitative comparison with neurophysiological data to back the authors interpretation.


Minor :

Line 100 : f is not defined at this point. The first time it is defined is line 118. Specifying f is an activation function when you use it, would improve the understanding of the article.

Line 129 : with respect $z^{(l+1)}$ —> with respect to $z^{(l+1)}$

[1] : Rao, Rajesh PN, and Dana H. Ballard. "Predictive coding in the visual cortex: a functional interpretation of some extra-classical receptive-field effects." Nature neuroscience 2.1 (1999): 79-87.

**Limitations:**

In general, I think have properly discussed the limitations of their work in their article.

**Strengths And Weaknesses:**

This article is clear and well written. It is easy to follow and solve most of the ’neuro-plausibility’ issue raised by the PC algorithm. This article is mainly grounded in theory (which is 100% fine), but in the same time it do not propose a huge theoretical breakthrough. On my opinion, the 2 main drawbacks of this article are 1) The theoretical part is not rigorous enough, 2) the comparison with neurophysiology is interesting but it more a matter of interpretation than a rigorous quantitative comparison with neurophysiological data (see section "Questions" for more information).

---

> ### Author Response · Authors · 2022-08-02
> **Response to reviewer 9q1E (2/2)**
>
> > * ($||z^{(l)}||^2$ as a ‘prior’) ‘Prior’ has a probabilist connotation, and here it doesn’t fit into your probabilist description of the model (nowhere, you have exhibited a normal prior for every  in section 2.1). You can easily circumvent that by calling the term  a ‘regularization term’ which has no probabilist connotation (I know I am being picky, but as this article is almost fully theoretical, it should be as rigorous as possible)
>
> We agree with this comment. In this manuscript, we were trying to point out that it can be interpreted as an *added* term in the prior. We have clarified this and thank the reviewer for pointing it out.
>
> >* (Clarification on experimental comparisons) It seems that the authors are comparing the training loss of PC, PC + cons. Why don’t you show the testing accuracy / testing loss ?
>
> In  all cases, we only show the validation loss and not the training loss. We will clarify this in the manuscript. We do not plot accuracy because we believe accuracy values in our experiments are generally not very informative in the linear setting (especially when applied to problems that are not linearly separable). In short, they overall resemble the same lines and we thought they clutter the paper. If the reviewer believes that these would be informative, we can add them back in. (See below for generalization)
>
> > * (Whitening and improvement in generalization) As the authors already mention (line 141), the whitening regularization added in the CCPC should improve the generalization abilities. But with the training loss only, how is it possible to assess the generalization gap (different between training and testing ? )
>
> This is a very good point. We agree with the suggestion and have looked into this. However:
>
> 1. The focus of the paper is to demonstrate that contrary to the belief of some recent work (e.g. [3]), predictive coding can indeed be made bio-plausible. The addition of the whitening was *motivated* by work that finds decorrelation improves generalization, but this improvement itself is not the focus of our work.
> 2. Unfortunately, the improvements in generalization performance in the linear setting are negligible. This can intuitively be understood if we think about linearity itself as an even stronger (arguably too strong) regularizer.  True to this, in all of our experiments, training performance is statistically the same as the validation performance. We have added figures to the supplementary materials section to demonstrate this (Sec. D)
>
> We will clarify these points in the manuscript.
>
> > * (Quantitative comparison with neurophysiological data.) The authors suggest that their CCPC allows to interpret the variable in terms of neurophysiology. Then I would suggest to conduct quantitative comparison with neurophysiological data to back the authors interpretation.
>
> Sadly, quantitative neurophysiological data of the type required to go beyond the analysis present in the paper is prohibitively challenging and as far as we are aware not currently available. However, the available anatomical data from the wiring pattern and qualitative data regarding the learning in pyramidal neurons are strong enough to lead many researchers to rule out the validity of Predictive Coding as a computational framework for the brain (e.g. [3]). In contrast, our algorithm is not only compatible with this data, but our theory also explains a number of these observations, e.g., the fact that the calcium plateau is a teaching signal is explained by Prop. 1.
>
>   Our hope is that by relating our work which is grounded in theory to the latest available experimental findings, we can motivate further dialog between theory and experiment in neuroscience and inspire experimentalists to perform experiments that can verify our predictions.
>
> ---
> [1] Beren Millidge, Anil Seth, and Christopher L Buckley. Predictive coding: a theoretical and experimental review, 2021.
>
> [2] James C.R. Whittington and Rafal Bogacz. An approximation of the error backpropagation algorithm in a predictive coding network with local hebbian synaptic plasticity. Neural Computation,
> 308 29(5):1229–1262, may 2017.
>
> [3] Micha Heilbron and Maria Chait. Great Expectations: Is there Evidence for Predictive Coding in Auditory Cortex? Neuroscience, 389:54–73, 2018.

---

> > ### Comment · Reviewer_9q1E · 2022-08-08
> > **Reponse to authors**
> >
> > I appreciate the clarification that have been made to the article, it is now (a bit) more rigorous. I still think this article would deserve more rigor to make it more impactful. I also think the additional experiments are making this article better. I have then increased my score from 5 to 6.

---

> ### Author Response · Authors · 2022-08-02
> **Response to reviewer 9q1E (1/2)**
>
> We are grateful to the reviewer for their in-depth review and detailed feedback on the derivation. We are pleased that the reviewer finds that our approach addresses most of the neuro-plausibility issues of the PC algorithm, as this is the primary goal of the paper. We have significantly increased our numerical experiments section and made modifications according to the suggestions of the reviewer.
>
> > * Theoretical part is not rigorous enough.
>
> We have added intermediate steps and filled in details to make the arguments more rigorous (see below).
>
> > * Factorization and the form of the hypothesis.
> >
> As the reviewer correctly points out, our assumption is a Markovian process. We have updated the draft to clarify this (above Eq. (1)).
>
>    However, note that while the unsupervised/generative setting is perhaps the most intuitive way to think about predictive coding (as in Rao and Ballard), supervised predictive coding is indeed a second major paradigm of the predictive coding framework (see for example [1]). The contents of Sec. 2 of the submission are a review of Whittington \& Bogacz [2] (Reference [4] in the submission), discussing supervised predictive coding, which is the basis for a number of recent follow-up work. Our contributions are deriving an upper bound and a more bio-realistic objective, which takes away some biologically unappealing features (e.g., symmetric connections) of the Whittington-Bogacz scheme. Also note that our approach can be similarly applied to the classical case of Rao and Ballard. We will clarify both of these facts in the manuscript. (Please note that to be more in line with standard ML notation, the order of layers in our submission is reversed compared to [2].)
>
> > * (Splitting $W$ into $W_a$ and $W_b$) The authors suggest that splitting $W$ into $W_a$  and $W_b$, doesn’t change the optimization problem. This specific point should be demonstrated.
>
> We agree that this is not very clear in the manuscript. We have added some more detail in the manuscript (footnote in Sec. 3.1). In short, the inequality follows because in Eq. 6, the arg-min of the objective $\hat L$ w.r.t. $W_a$ is equal to the arg-min of the objective $\hat L$ w.r.t. $W_b$ (with fixed $Z$), and also equal to the arg-min of the original objective $L$ w.r.t. $W$. This is because the objective (with fixed Z) is convex in the $W$'s. We can therefore optimize for $W_a$ and $W_b$, plug it back into the objective $\hat L$ and verify directly that the equality holds, i.e., $\min_W L(W,Z) = \min_{W_a,W_b} \hat L(W_a, W_b,Z)$ explicitly., where $L$ is the original objective and $\hat L$ is the objective with $W_a$ and $W_b$ introduced.
>
> Of course, as the reviewer correctly points out, after the next step (i.e., in Eq. 8) $W_a$ and $W_b$ are no longer treated the same, and the objective is no longer equal to the original one. However, with this order of steps, we arrive at a consistent upper bound for the original objective. That is, in the first step, we introduce $W_a$ and $W_b$ and we have equality. In the following steps, we treat them differently and arrive at upper bounds.
>
> > * (Inequality in Eq.8.) In the right term of Eq. 8 there is a minimization w.r.t.  $W_b^{(l)}$, but as expected just after this equation,  $L^{(l)}$ does not depend on  $W_b^{(l)}$  but rather on  $W_b^{(l-1)}$. Is it a typo in your inequality in Eq. 8? Even if it is a typo, this inequality would deserve more explanation.
>
> Thank you for pointing out the lack of clarity and  the typo. We have added details in the derivation below equation 8. In short, the inequality follows from the fact that the right hand side is an evaluation of the same function $\hat L$, at a point that is not given by the minimum. This is because $\sum_l \hat L^{(l)} = \hat L$ (which is what we meant by consistency). In this sense, the inequality is trivial: if one evaluates a function at any point (regardless of how this point is chosen), one arrives at an upper bound for the minimum of the function. Another way to see this is that the splitting in Eq. 8 is equivalent to adding a number of stop-gradients (for $Z$'s) to $\hat L$. The minimum of any function will always be smaller than (or at most equal to) any value achieved when performing gradient descent with stop-gradients. We provide a more rigorous (but longer) derivation in the supplementary materials.

---

### Official Review · Reviewer_UwcZ · 2022-07-11

**Rating:** 7
**Confidence:** 3
**Soundness:** 3 good
**Presentation:** 3 good
**Contribution:** 3 good

**Summary:**

This paper proposes a variant of predictive coding (a biologically inspired framework for neural computation and learning) that better aligns with the known physiology and anatomy of cortex. This is achieved by applying a decorrelation constraint, yielding an upper bound on the PC objective. The resulting architecture is proposed to map onto the cortical microcircuitry, with each of the 3 main terms affecting a separate compartment of the pyramidal neurons. Numerical experiments show that this BioCCPC model can approach the performance of vanilla PC on MNIST.

**Questions:**

* Some of the mathematical derivations seem to skip a number of steps, making them difficult to follow. A clear example is Equation (9), whose form is very difficult to relate to any of the previous equations. Similarly, Equation (4) doesn’t naturally follow from the previous ones.
* Should the $v_a$ and $n$ updates be switched in Algorithm 1? $v_a$ is updated using $n$, which isn’t assigned until the next step…
* Line 244, should it be the optimal linear inferred value for $y_{T+1}$ instead of $y_T$ ?


**Limitations:**

The main limitation due to the linearity assumption is properly acknowledged.

**Strengths And Weaknesses:**

Strengths:
* Potentially important variant of predictive coding that could be of interest to neuroscientists and ML scientists.
* Clear biological motivation
* Thorough theoretical framework

Weaknesses:
* All results are obtained under an assumption of linearity that could limit the usefulness of the findings in ML applications
* The theoretical derivations are not always easy to follow
* Experiments are restricted to a toy model and dataset (MNIST).

---

> ### Author Response · Authors · 2022-08-02
> **Response to reviewer UwcZ**
>
> We are grateful to the reviewer for their thoughtful comments and for pointing out the issues in the derivation and the algorithm. We are encouraged by the reviewer's belief in the possible importance of our submission in both neuroscience and ML. We took the reviewer's comments and suggestions and made the following modifications and additions:
>
> > * Experiments are restricted to a toy model and dataset (MNIST).
>
> In the submission, we tested the algorithm on MNIST and, in the supplementary materials, provided results for MediaMill (a multi-view dataset of video frame/annotation pairs). We have now expanded our empirical verifications to include several new experiments. In order to show that our biologically realistic algorithm performs as well as the previous non-realistic algorithm, we performed experiments on a number of other datasets: Fashion MNIST (FMNIST), CIFAR10, CIFAR100, and Labeled Faced in the Wild (LFW). The FMNIST and CIFAR experiments were done in the supervised setting. However, the experiment on the LFW dataset, which includes pairs of faces from the same person, was performed in a self-supervised fashion with an auto-encoder structure. In all tested scenarios (MNIST, FMNIST, CIFAR10, CIFAR100, MediaMill, LFW), Bio-PCN achieves the same loss as PCN, demonstrating that optimizing the upper bound that we derived effectively optimizes the original objective. The new experimental results can be found in the updated manuscript  (supplementary materials Sec. C.2)
>
>   Also note that in general, training a narrow architecture with a bottleneck (as we have in our networks) is more difficult than training a wide overparametrized network. This is in particular true for linear networks and especially for training algorithms that do not perform backpropagation. In this sense, we performed the experiments that we believed would be most challenging for our algorithm.  We will add a paragraph explaining this in the experiment section.
>
> > * The theoretical derivations are not always easy to follow.
> >  * Some of the mathematical derivations seem to skip a number of steps, making them difficult to follow. A clear example is Equation (9), whose form is very difficult to relate to any of the previous equations. Similarly, Equation (4) doesn’t naturally follow from the previous ones.
>
> Thank you for pointing this out.  We have added multiple intermediate steps (respectively in and below equation (4), and above equation (9)) that we hope clarify the derivation.
>
> > * All results are obtained under an assumption of linearity that could limit the usefulness of the findings in ML applications
>
>  We agree with this assessment. However, theoretical proofs and derivations as in this work are very challenging without simplifying assumptions such as linearity. Despite its limitations,  the linear setting has proved useful, in the past, for understanding many features of deep learning practice [1]. We hope that our results (for example the relation to neurophysiology and Prop. 1 regarding how the error signal is implicitly computed without the need for backpropagation) will also shed some light in the general setting.
>
> > * Should the $v_a$ and $n$ updates be switched in Algorithm 1?  $v_a$ is updated using $n$, which isn’t assigned until the next step…
> > * Line 244, should it be the optimal linear inferred value for $y_{T+1}$ instead of $y_T$ ?
>
> You are correct on both counts. Thank you for pointing these out, we have corrected these in the manuscript.
>
> ---
> [1] Roberts, Daniel A., Sho Yaida, and Boris Hanin. The Principles of Deep Learning Theory: An Effective Theory Approach to Understanding Neural Networks. Cambridge University Press, 2022.

---

> > ### Comment · Reviewer_UwcZ · 2022-08-06
> > **Thanks for the authors response**
> >
> > I appreciate the new numerical experiments (e.g. CIFAR), as well as the improved mathematical derivations. The paper has improved, and I will (slightly) improve my original score accordingly.
> > In my opinion, there is still a lot of work needed to adapt this method to (non-linear) settings that will prove practically useful in ML applications. Nonetheless, the present study can already be viewed as a useful initial theoretical contribution to bridge neuroscience knowledge with ML objectives.

---

### Author Response · Authors · 2022-08-02
**General response**

We thank the reviewers for their thoughtful feedback. We are encouraged that they found our approach clearly biologically motivated [R1] and our algorithm an important variant of predictive coding [R1], which addresses many of the neuro-plausibility issues of the PC algorithm [R2]. We are also pleased that the reviewers generally find our theoretical analysis thorough and sound [R3].

We found the reviewer comments very constructive, and in accordance with their suggestions, have added new experiments and made modifications. We summarize the changes and additions here and provide more detailed responses to each reviewer individually.

* **New numerical experiments.** We expanded our numerical experiments section by performing a number of new simulations. Whereas we initially tested on MNIST and MediaMill, we have now also verified our algorithm on Fashion-MNIST, CIFAR10, CIFAR100, and LFW (Labeled Faces in the Wild). Furthermore, whereas in the submission, we primarily looked at learning in the supervised setting with labeled datasets, we have now more experiments in the self-supervised setting with a symmetric auto-encoder-type architecture (MediaMill and LFW). In all tested scenarios (MNIST, FMNIST, CIFAR10, CIFAR100, MediaMill, LFW), Bio-PCN achieves the same loss as PCN, demonstrating that optimizing the upper bound that we derived effectively optimizes the original objective.  The new experimental results can be found in the updated manuscript (supplementary materials Sec. C.2)

* **Added details to theoretical arguments.** We add further details that fill in the steps in our derivation. We also add clarifications which we believe help the rigor of the arguments.

* **Clarified novelty and contributions.** We apologize for the lack of clarity on this matter. The main contribution of this work is the demonstration that the predictive coding framework is compatible with (and a good fit for) the known neurophysiology of the brain (in terms of anatomical connectivity, neural dynamics, and learning rules). We demonstrated this by deriving a novel upper bound for the predictive coding loss and theoretically proving several properties of the circuit. We believe this is relevant to the community because the previous implementations of predictive coding require connectivity patterns that are not experimentally observed, to the point that many researchers have questioned the validity of PC as a computational framework for the brain [1].

    We will clarify this and add a new section clarifying prior work.

___
[1] Micha Heilbron and Maria Chait. Great Expectations: Is there Evidence for Predictive Coding in Auditory Cortex? Neuroscience, 389:54–73, 2018.

---

### Meta-Review · Area_Chair_DDQt · 2022-08-25

**Recommendation:** Accept
**Confidence:** Less certain

**Metareview:**

A biologically more plausible extension of predictive coding that incorporates physiological detail is presented in this manuscript. I appreciate all discussions and extra experiments and improvements made to the paper. I agree with one of the reviewers who noted that it's a "potentially important variant of predictive coding that could be of interest to neuroscientists and ML scientists." Considering the theoretical contributions and new formulation to a subfield with significant growing interest, I recommend its acceptance to NeurIPS.

**Award:**

No

---

### Decision · Program_Chairs · 2022-09-14

Accept